# Approximate bi-criteria search by efficient representation of subsets of the Pareto-optimal frontier

**Oren Salzman**[†]
[†]Technion, Israel Institute of Technology
Haifa 32000, Israel
salzman@cs.technion.ac.il

## Abstract

We consider the bi-criteria shortest-path problem where we want to compute shortest paths on a graph that simultaneously balance two cost functions. While this problem has numerous applications, there is usually no path minimizing both cost functions simultaneously. Thus, we typically consider the set of paths where no path is strictly better than the others in both cost functions, a set called the Pareto-optimal frontier. Unfortunately, the size of this set may be exponential in the number of graph vertices and the general problem is NP-hard. While existing schemes to approximate this set exist, they may be slower than exact approaches when applied to relatively small instances and running them on graphs with even a moderate number of nodes is often impractical. The crux of the problem lies in how to efficiently approximate the Pareto-optimal frontier. Our key insight is that the Pareto-optimal frontier can be approximated using *pairs* of paths. This simple observation allows us to run a best-first-search while efficiently and effectively pruning away intermediate solutions in order to obtain an approximation of the Pareto frontier for any given approximation factor. We compared our approach with an adaptation of $\mathsf{BOA}^*$, the state-of-the-art algorithm for computing exact solutions to the bi-criteria shortest-path problem. Our experiments show that as the problem becomes harder, the speedup obtained becomes more pronounced. Specifically, on large roadmaps, when using an approximation factor of $10\%$ we obtain a speedup on the average running time of more than $\times 19$.

## 1   Introduction & Related Work

We consider the bi-criteria shortest-path problem, an extension to the classical (single-criteria) shortest-path problem where we are given a graph $G = (V, E)$ and each edge has two cost functions. Here, we are required to compute paths that balance between the two cost functions. The well-studied problem (Chinchuluun and Pardalos 2007) has numerous applications. For example, given a road network, the two cost functions can represent travel times and distances and we may need to consider the set of paths that allow to balance between these costs. Other applications include planning of power-transmission lines (Bachmann et al. 2018) and planning how to transport hazardous material in order to balance between minimizing the travel distance and the risk of exposure for residents (Bronfman et al. 2015).

There usually is no path minimizing all cost functions simultaneously. Thus, we typically consider the set of paths where no path is strictly better then the others for both cost functions, a set called the *Pareto-optimal frontier*. Unfortunately, the problem is NP-hard (Serafini 1987) as the cardinality of the size of the Pareto-optimal frontier may be exponential in $|V|$ (Ehrgott 2005; Breugem, Dollevoet, and van den Heuvel 2017) and even determining whether a path belongs to the Pareto-optimal frontier is NP-hard (Papadimitriou and Yannakakis 2000).

Existing methods either try to efficiently compute the Pareto-optimal frontier or to relax the problem and only compute an approximation of this set.

**Efficient computation of the Pareto-optimal frontier.** To efficiently compute the Pareto-optimal frontier, adaptations of the celebrated $\mathsf{A}^*$ algorithm (Hart, Nilsson, and Raphael 1968) were suggested. Stewart et al. (1991) introduced Multi-Objective $\mathsf{A}^*$ ($\mathsf{MOA}^*$) which is a multiobjective extension of $\mathsf{A}^*$. The most notable difference between $\mathsf{MOA}^*$ and $\mathsf{A}^*$ is in maintaining the Pareto-optimal frontier to intermediate vertices. This requires to check if a path $\pi$ is *dominated* by another path $\tilde{\pi}$. Namely, if both of $\tilde{\pi}$'s costs are smaller than $\pi$'s costs. As these dominance checks are repeatedly performed, the time complexity of the checks play a crucial role for the efficiency of such bi-criteria shortest-path algorithms. $\mathsf{MOA}^*$ was later revised (Mandow and De La Cruz 2005; Mandow and De La Cruz 2010; Pulido, Mandow, and Pérez-de-la Cruz 2015) with the most efficient variation, termed bi-Objective $\mathsf{A}^*$ ($\mathsf{BOA}^*$) (Ulloay et al. 2020) allowing to compute these operations in $O(1)$ time when a consistent heuristic is used.[1]

**Approximating the Pareto-optimal frontier.** Initial methods in computing an approximation of the Pareto-optimal frontier were directed towards devising a Fully Polynomial Time Approximation Scheme[2]

---

[1]A heuristic function is said to be consistent if its estimate is always less than or equal to the estimated distance from any neighbouring vertex to the goal, plus the cost of reaching that neighbour.

[2]An FPTAS is an approximation scheme whose time complexity is polynomial in the input size and also polynomial in $1/\varepsilon$

(FPTAS) (Vazirani 2001). Warburton (1987) proposed a method for finding an approximate Pareto optimal solution to the problem for any degree of accuracy using scaling and rounding techniques. Perny and Spanjaard (2008) presented another FPTAS given that a finite upper bound $L$ on the numbers of arcs of all solution-paths in the Pareto-frontier is known. This requirement was later relaxed (Tsaggouris and Zaroliagis 2009; Breugem, Dollevoet, and van den Heuvel 2017) by partitioning the space of solutions into cells according to the approximation factor and, roughly speaking, taking only one solution in each grid cell. Unfortunately, the running times of FPTASs are typically polynomials of high degree, and hence they may be slower than exact approaches when applied to relatively-small instances and running them on graphs with even a moderate number of nodes (e.g., $\approx 10,000$) is often impractical (Breugem, Dollevoet, and van den Heuvel 2017).

A different approach to compute a subset of the Pareto-optimal solution is to find all extreme supported non-dominated points (i.e., the extreme points on the convex hull of the Pareto-optimal set) (Sedeno-Noda and Raith 2015). Taking a different approach Legriel et al. (2010) suggest a method based on satisfiability/constraint solvers. Alternatively, a simple variation of $\mathsf{MOA}^*$, termed $\mathsf{MOA}^*_\varepsilon$ allows to compute an approximation of the Pareto-optimal frontier by prunning intermediate paths that are approximately dominated by already-computed solutions (Perny and Spanjaard 2008). However, as we will see, this allows to prune only a small subset of paths that may be pruned.

Finally, recent work (Bökler and Chimani 2020) conducts a comprehensive computational study with an emphasis on multiple criteria. Similar to the aforementioned FPTASs, their framework still partitions the space prior to running the algorithm.

**Key contribution.** To summarize, exact methods compute a solution set whose size is often exponential in the size of the input. While one would expect that approximation algorithms will allow to dramatically speed up computation times, in practice their running times are often slower than exact solutions for FPTAS's because they partition the space of solution into cells according to the approximation factor in advance. Alternative methods only prune paths that are approximately dominated by already-computed solutions.

Our key insight is that we can efficiently partition the space of solution into cells during the algorithm's execution (and not a-priori). This allows us to efficiently and effectively prune away intermediate solutions in order to obtain an approximation of the Pareto frontier for any given approximation factor $\varepsilon$ (this will be formalized in Sec. 2). This is achieved by running a best-first search on *path pairs* and not individual paths. Such path pairs represent a subset of the Pareto frontier such that any solution in this subset is approximately dominated by the two paths. Using concepts that draw inspiration from a recent search algorithm from the robotics literature (Fu et al. 2019), we propose Path-Pair

____________________

where $\varepsilon$ is the approximation factor.

$\mathsf{A}^*$ ($\mathsf{PP\text{-}A}^*$). $\mathsf{PP\text{-}A}^*$ dramatically reduces the computational complexity of the best-first search by merging path pairs while still ensuring that an approximation of the Pareto-optimal frontier is obtained for any desired approximation.

For example, on a roadmap of roughly 1.5 million vertices, $\mathsf{PP\text{-}A}^*$ approximates the Pareto optimal frontier within a factor of $1\%$ in roughly 13 seconds on average on a commodity laptop. We compared our approach with an adaptation of $\mathsf{BOA}^*$ (Ulloay et al. 2020), the state-of-the-art algorithm for computing exact solutions to the bi-criteria shortest-path problem, which we term $\mathsf{BOA}^*_\varepsilon$. $\mathsf{BOA}^*_\varepsilon$ computes near optimal solutions by using the approach suggested in (Perny and Spanjaard 2008). Our experiments show that as the problem becomes harder, the speedup that $\mathsf{PP\text{-}A}^*$ may offer becomes more pronounced. Specifically, on the aforementioned roadmap and using an approximation factor of $10\%$, we obtain a speedup on the average running time of more than $\times 19$ and a maximal speedup of over $\times 25$.

## 2 Problem definition

Let $G = (V, E)$ be a graph, $c_1 : E \to \mathbb{R}$ and $c_2 : E \to \mathbb{R}$ be two cost functions defined over the graph edges. A path $\pi = v_1, \ldots v_k$ is a sequence of vertices where consecutive vertices are connected by an edge. We extend the two cost functions to paths as follows:

$$c_1(\pi) = \sum_{i=1}^{k-1} c_1(v_i, v_{i+1}) \quad \text{and} \quad c_2(\pi) = \sum_{i=1}^{k-1} c_2(v_i, v_{i+1}).$$

Unless stated otherwise, all paths start at the same specific vertex $v_{\text{start}}$ and $\pi_u$ will denote a path to vertex $u$.

**Definition 1** (Dominance). *d We say that $\pi_u$* strictly dominates *$\tilde{\pi}_u$ if (i) $\pi_u$ weakly dominates $\tilde{\pi}_u$ and (ii) $c_1(\pi_u) < c_1(\tilde{\pi}_u)$ or $c_2(\pi_u) < c_2(\tilde{\pi}_u)$.*

**Definition 2** (Approximate dominance). *Let $\pi_u$ and $\tilde{\pi}_u$ be two paths to vertex $u$ and let $\varepsilon_1 \geq 0$ and $\varepsilon_2 \geq 0$ be two real values. We say that $\pi_u$ $(\varepsilon_1, \varepsilon_2)$-dominates $\tilde{\pi}_u$ if (i) $c_1(\pi_u) \leq (1 + \varepsilon_1) \cdot c_1(\tilde{\pi}_u)$ and (ii) $c_2(\pi_u) \leq (1 + \varepsilon_2) \cdot c_2(\tilde{\pi}_u)$. When $\varepsilon_1 = \varepsilon_2$, we will sometimes say that $\pi_u$ $(\varepsilon_1)$-dominates $\tilde{\pi}_u$ and call $\varepsilon_1$ the* approximation factor.

**Definition 3** ((approximate) Pareto-optimal frontier). *The* Pareto-optimal frontier *$\Pi_u$ of a vertex $u$ is a set of paths connecting $v_{\text{start}}$ and $u$ such that (i) no path in $\Pi_u$ is strictly dominated by any other path from $v_{\text{start}}$ to $u$ and (ii) every path from $v_{\text{start}}$ to $u$ is weakly dominated by a path in $\Pi_u$. Similarly, for $\varepsilon_1 \geq 0$ and $\varepsilon_2 \geq 0$ the* approximate Pareto optimal frontier[3] *$\Pi_u(\varepsilon_1, \varepsilon_2) \subseteq \Pi_u$ is a subset of $u$'s Pareto frontier such that every path in $\Pi_u$ is $(\varepsilon_1, \varepsilon_2)$-dominated by a path in $\Pi_u(\varepsilon_1, \varepsilon_2)$.*

For brevity we will use the terms (approximate) Pareto frontier to refer to the (approximate) Pareto optimal frontier. For a visualization of these notions, see Fig. 1.

We are now ready to formally define our search problems.

____________________

[3]Our definition of an approximate Pareto optimal frontier slightly differs from existing definitions (Breugem, Dollevoet, and van den Heuvel 2017) which do not require that the approximate Pareto frontier is a subset of the Pareto-optimal frontier.

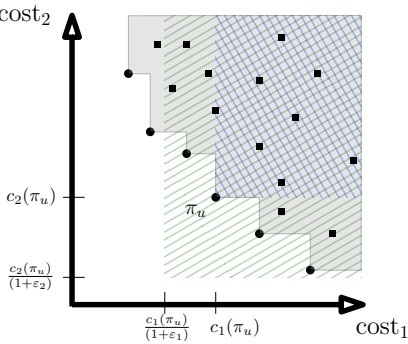

Figure 1: Dominance, approximate dominance and Pareto frontier. Given start and target vertices, we consider each path $\pi_u$ as a 2D point $(c_1(\pi_u), c_2(\pi_u))$ according to the two cost functions (points and squares). The set of all possible paths dominated and approximately dominated by path $\pi_u$ are depicted in blue and green, respectively. The Pareto frontier $\Pi_u$ is the set of all black points that collectively dominate all other possible paths (squares in grey region).

**Problem 1** (Bi-criteria shortest path). *Let $G$ be a graph, $c_1, c_2 : E \rightarrow \mathbb{R}$ two cost functions and $v_{\text{start}}$ and $v_{\text{goal}}$ be start and goal vertices, respectively. The* bi-criteria shortest path *problem calls for computing the Pareto frontier $\Pi_{v_{\text{goal}}}$.*

**Problem 2** (Bi-criteria approximate shortest path). *Let $G$ be a graph, $c_1, c_2 : E \rightarrow \mathbb{R}$ two cost functions and $v_{\text{start}}$ and $v_{\text{goal}}$ be start and goal vertices, respectively. Given $\varepsilon_1 \geq 0$ and $\varepsilon_2 \geq 0$, the* bi-criteria approximate shortest path *problem calls for computing an approximate Pareto frontier $\Pi_{v_{\text{goal}}}(\varepsilon_1, \varepsilon_2)$.*

## 3 Algorithmic Background

In this section we describe two approaches to solve the bi-criteria shortest-path problem (Problem 1). With the risk of being tedious, we start with a brief review of best-first search algorithms as both state-of-the-art bi-criteria shortest path algorithms, as well as ours, rely heavily on this algorithmic framework. We note that the description of best-first search we present here can be optimized but this version will allow us to better explain the more advanced algorithms.

A best-first search algorithm (Alg. 1) computes a shortest path from $v_{\text{start}}$ to $v_{\text{goal}}$ by maintaining a priority queue, called an OPEN list, that contains all the nodes that have not been expanded yet (line 1). Each node is associated with a path $\pi_u$ from $v_{\text{start}}$ to some vertex $u \in V$ (by a slight abuse of notation we will use paths and nodes interchangeably which will simplify algorithm's descriptions in the next sections). This queue is ordered according to some cost function called the $f$-value of the node. For example, in Dijkstra and $\mathsf{A}^*$, this is the computed cost from $v_{\text{start}}$ (also called its $g$-value) and the computed cost from $v_{\text{start}}$ added to the heuristic estimate to reach $v_{\text{goal}}$, respectively.

At each iteration (lines 3-13), the algorithm extracts the most-promising node from OPEN (line 3), checks if it has the potential to be a better solution than any found so far

---

**Algorithm 1** Best First Search

**Input:** $(G = (V, E), v_{\text{start}}, v_{\text{goal}}, \ldots)$
1: OPEN $\leftarrow$ new node $\pi_{v_{\text{start}}}$
2: **while** OPEN $\neq \emptyset$ **do**
3:     $\pi_u \leftarrow$ OPEN.**extract_min**()
4:     **if is_dominated**$(\pi_u)$ **then**
5:       **continue**
6:     **if** $u = v_{\text{goal}}$ **then**       ▷ reached goal
7:       **merge_to_solutions**$(\pi_u, \text{solutions})$
8:       **continue**
9:     **for** $e = (u, v) \in$ neighbors$(u, G)$ **do**
10:       $\pi_v \leftarrow$ **extend**$(\pi_u, e)$
11:       **if is_dominated**$(\pi_v)$ **then**
12:         **continue**
13:       **insert**$(\pi_v, \text{OPEN})$
14: **return** all extreme paths in solutions

---

(line 4). If this is the case and we reached $v_{\text{goal}}$, the solution set is updated (in single-criteria shortest path, once a solution is found, the search can be terminated). If not, we extend the path represented by this node to each of it's neighbors (line 10). Again, we check if it has the potential to be a better solution than any found so far (line 11). If this is the case, it is added to the OPEN list.

Different single-criteria search algorithms such as Dijkstra, $\mathsf{A}^*$, $\mathsf{A}^*_\varepsilon$ as well as bi-criteria search algorithms such BOA$^*$ fall under this framework. They differ with how OPEN is ordered and how the different functions (highlighted in Alg. 1) are implemented.

**Bi-Objective $\mathsf{A}^*$ (BOA$^*$)** To efficiently solve Problem 1, bi-Objective $\mathsf{A}^*$ (BOA$^*$) runs a best-first search. The algorithm is endowed with two heursitic functions $h_1, h_2$ estimating the cost to reach $v_{\text{goal}}$ from any vertex according to $c_1$ and $c_2$, respectively. Here, we assume that these heuristic functions are admissible and consistent. This is key as the efficiency of BOA$^*$ relies on this assumption.

Given a node $\pi_u$, we define $g_i(\pi_u)$ to be the computed distance according to $c_i$. It can be easily shown that in best-first search algorithms $g_i := c_i(\pi_u)$. Additionally, we define $f_i(\pi_u) := g_i(\pi_u) + h_i(\pi_u)$. Although the cost and the $g$-value of a path can be used interchangeably, we will use the former to describe general properties of paths and the later to describe algorithm operations. Nodes in OPEN are ordered lexicographically according to $(f_1, f_2)$ which concludes the description of how **extract_min** and **insert** (lines 3 and 13, respectively) are implemented.

Domination checks, which are typically time-consuming in bi-criteria search algorithms are implemented in $O(1)$ per node by maintaining for each vertex $u \in V$ the minimal cost to reach $u$ according to $c_2$ computed so for. This value is maintained in a map $g_2^{\min} : V \rightarrow \mathbb{R}$ which is initialized to $\infty$ for each vertex. This allows to implement the function **is_dominated** for a node $\pi_u$ by testing if

$$g_2(\pi_u) \geq g_2^{\min}(u) \text{ or } f_2(\pi_u) \geq g_2^{\min}(v_{\text{goal}}). \quad (1)$$

The first test checks if the node is dominated by an already-extended node and replaces the CLOSED list typically used in $A^*$-like algorithms. The second test checks if the node has the potential to reach the goal with a solution whose cost is not dominated by any existing solution. Finally, the function **merge_to_solutions** simply adds a newly-found solution to the solution set.

**Computing the approximate Pareto frontier**   Perny and Spanjaard (2008) suggest to compute an approximate Pareto frontier by endowing the algorithm with an approximation factor $\varepsilon$. When a node is popped from OPEN, we test if its $f$-value is $\varepsilon$-dominated by any solution that was already computed. While this algorithm was presented before $BOA^*$ and hence uses computationally-complex dominance checks, we can easily use this approach to adapt $BOA^*$ to compute an approximate Pareto frontier. This is done by replacing the dominance check in Eq. 1 with the test

$$g_2(\pi_u) \geq g_2^{\min}(v) \text{ or } (1+\varepsilon) \cdot f_2(\pi_u) \geq g_2^{\min}(v_{\text{goal}}). \quad (2)$$

We call this algorithm $BOA_\varepsilon^*$.

# 4   Algorithmic Framework

## 4.1   Preliminaries

Recall that (single-criteria) shortest-path algorithms such as $A^*$ find a solution by computing the shortest path to all nodes that have the potential to be on the shortest path to the goal (namely, whose $f$-value is less than the current estimate of the cost to reach $v_{\text{goal}}$). Similarly, bi-criteria search algorithms typically compute for each node the subset of the Pareto frontier that has the potential to be in $\Pi_{v_{\text{goal}}}$.

Now, near-optimal (single-criteria) shortest-path algorithms such as $A_\varepsilon^*$ (Pearl and Kim 1982) attempt to speed this process by only *approximating* the shortest path to intermediate nodes. Similarly, we suggest to construct only an approximate Pareto frontier for intermediate nodes which, in turn, will allow to dramatically reduce computation times. Looking at Fig. 1, one may suggest to run an $A^*$-like search and if a path $\pi_u$ on the Pareto frontier $\Pi_u$ of $u$ is approximately dominated by another path $\tilde{\pi}_u \in \Pi_u$, then discard $\pi_u$. Unfortunately, this does not account for paths in $\Pi_u$ that may have been approximately dominated by $\pi_u$ and hence discarded in previous iterations of the search. Existing methods use very conservative bounds to prune intermediate paths. For example, as stated in Sec. 1, if a bound $L$ on the length of the longest path exists, we can use this strategy by replacing $(1+\varepsilon)$ with $(1+\varepsilon)^{1/L}$ to account for error propagation (Perny and Spanjaard 2008).

In contrast, we suggest a simple-yet-effective method to prune away approximately-dominated solutions using the notion of a partial Pareto frontier which we now define.

**Definition 4** (Partial Pareto frontier PPF). *Let $\pi_u^{tl}, \pi_u^{br} \in \Pi_u$ be two paths on the Pareto frontier of vertex $u$ such that $c_1(\pi_u^{tl}) < c_1(\pi_u^{br})$ (here, $\mathtt{tl}$ and $\mathtt{br}$ are shorthands for "top left" and "bottom right" for reasons which will soon be clear). Their* partial Pareto frontier $\mathrm{PPF}_u^{\pi_u^{tl}, \pi_u^{br}} \subseteq \Pi_u$ *is a subset of a Pareto frontier such that if $\pi_u \in \Pi_u$ and*

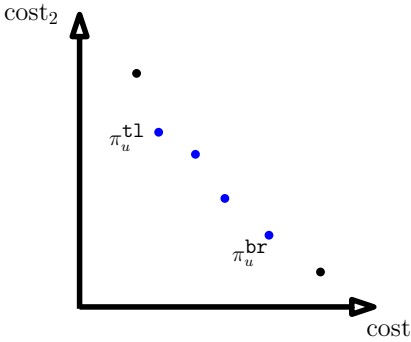

Figure 2: The partial Pareto frontier of two paths $\pi_u^{\mathtt{tl}}$ and $\pi_u^{\mathtt{br}}$ is the set of all paths (blue dots) on the Pareto frontier (blue and black dots) between these paths. Lemma 1 implies that any path represented by a blue dot is approximately dominated by $\pi_u^{\mathtt{tl}}$ and $\pi_u^{\mathtt{br}}$ for $\varepsilon_1 = \frac{c_1(\pi_u^{\mathtt{br}}) - c_1(\pi_u^{\mathtt{tl}})}{c_1(\pi_u^{\mathtt{tl}})}$ and $\varepsilon_2 = \frac{c_2(\pi_u^{\mathtt{tl}}) - c_2(\pi_u^{\mathtt{br}})}{c_2(\pi_u^{\mathtt{br}})}$.

$c_1(\pi_u^{tl}) < c_1(\pi_u) < c_1(\pi_u^{br})$ then $\pi_u \in \mathrm{PPF}_u^{\pi_u^{tl}, \pi_u^{br}}$. *The paths $\pi_u^{tl}, \pi_u^{br}$ are called the* extreme *paths of $\mathrm{PPF}_u^{\pi_u^{tl}, \pi_u^{br}}$ For a visualization, see Fig. 2.*

**Definition 5** (Bounded PPF). *A partial Pareto frontier $\mathrm{PPF}_u^{\pi_u^{tl}, \pi_u^{br}} \subseteq \Pi_u$ is $(\varepsilon_1, \varepsilon_2)$-bounded if*

$$\varepsilon_1 \geq \frac{c_1(\pi_u^{br}) - c_1(\pi_u^{tl})}{c_1(\pi_u^{tl})} \text{ and } \varepsilon_2 \geq \frac{c_2(\pi_u^{tl}) - c_2(\pi_u^{br})}{c_2(\pi_u^{br})}.$$

**Lemma 1.** *If $\mathrm{PPF}_u^{\pi_u^{tl}, \pi_u^{br}}$ is an $(\varepsilon_1, \varepsilon_2)$-bounded partial Pareto frontier then any path in $\mathrm{PPF}_u^{\pi_u^{tl}, \pi_u^{br}}$ is $(\varepsilon_1, \varepsilon_2)$-dominated by both $\pi_u^{tl}$ and $\pi_u^{br}$.*

*Proof.* Let $\pi_u \in \mathrm{PPF}_u^{\pi_u^{tl}, \pi_u^{br}}$. By definition, we have that $c_1(\pi_u^{tl}) < c_1(\pi_u)$ and that $\varepsilon_1 \geq \frac{c_1(\pi_u^{br}) - c_1(\pi_u^{tl})}{c_1(\pi_u^{tl})}$. Thus,

$$c_1(\pi_u^{br}) \leq (1 + \varepsilon_1) \cdot c_1(\pi_u^{tl}) < (1 + \varepsilon_1) \cdot c_1(\pi_u).$$

As $c_2(\pi_u^{br}) < c_2(\pi_u)$, we have that $\pi_u^{br}$ approximately dominates $\pi_u$.

Similarly, by definition, we have that $c_2(\pi_u) > c_2(\pi_u^{br})$ and that $\varepsilon_2 \geq \frac{c_2(\pi_u^{tl}) - c_2(\pi_u^{br})}{c_2(\pi_u^{br})}$. Thus,

$$c_2(\pi_u^{tl}) \leq (1 + \varepsilon_2) \cdot c_2(\pi_u^{br}) < (1 + \varepsilon_2) \cdot c_1(\pi_u).$$

As $c_1(\pi_u^{tl}) < c_1(\pi_u)$, we have that $\pi_u^{tl}$ approximately dominates $\pi_u$.  □

## 4.2   Algorithmic description

In contrast to standard search algorithms which incrementally construct shortest paths from $v_{\text{start}}$ to the graph vertices, our algorithm will incrementally construct $(\varepsilon_1, \varepsilon_2)$-bounded partial Pareto frontiers. Lemma 1 suggests a method to efficiently represent and maintain these frontiers for any approximation factors $\varepsilon_1$ and $\varepsilon_2$. Specifically, for a vertex $u$, $PP\text{-}A^*$ will maintain *path pairs* corresponding

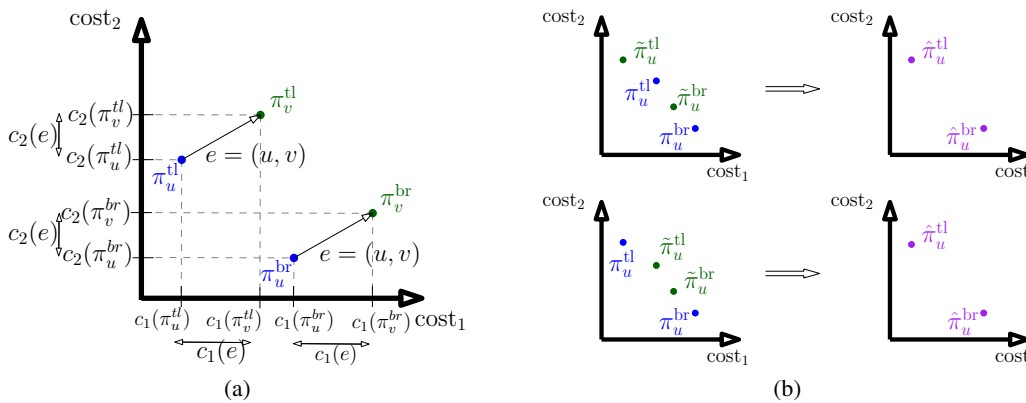

(a)

(b)

Figure 3: Operations on path pairs. (a) Extend operation. The path pair $(\pi_u^{\mathtt{tl}}, \pi_u^{\mathtt{br}})$ (blue) is extended by edge $e = (u, v)$ to obtain the path pair $(\pi_v^{\mathtt{tl}}, \pi_v^{\mathtt{br}})$ (green). (b) Merge operation. Two examples of merging the path pair $(\pi_u^{\mathtt{tl}}, \pi_u^{\mathtt{br}})$ (blue) with the path pair $(\tilde{\pi}_u^{\mathtt{tl}}, \tilde{\pi}_u^{\mathtt{br}})$ (green) to obtain the path pair $(\hat{\pi}_u^{\mathtt{tl}}, \hat{\pi}_u^{\mathtt{br}})$ (purple).

to the extreme paths in partial Pareto frontiers. For each path pair $(\pi_u^{\mathtt{tl}}, \pi_u^{\mathtt{br}})$ we have that $c_1(\pi_u^{\mathtt{tl}}) \le c_1(\pi_u^{\mathtt{br}})$ and $c_2(\pi_u^{\mathtt{tl}}) \ge c_2(\pi_u^{\mathtt{br}})$.

Before we explain how path pairs will be used let us define operations on path pairs: The first operation we consider is *extending* a path pair $(\pi_u^{\mathtt{tl}}, \pi_u^{\mathtt{br}})$ by an edge $e = (u, v)$, which simply corresponds to extending both $\pi_u^{\mathtt{tl}}$ and $\pi_u^{\mathtt{br}}$ by $e$. The second operation we consider is *merging* two path pairs $(\pi_u^{\mathtt{tl}}, \pi_u^{\mathtt{br}})$ and $(\tilde{\pi}_u^{\mathtt{tl}}, \tilde{\pi}_u^{\mathtt{br}})$. This operation constructs a new path pair $(\hat{\pi}_u^{\mathtt{tl}}, \hat{\pi}_u^{\mathtt{br}})$ such that

$$\hat{\pi}_u^{\mathtt{tl}} = \begin{cases} \pi_u^{\mathtt{tl}} & \text{if } c_1(\pi_u^{\mathtt{tl}}) \le c_1(\tilde{\pi}_u^{\mathtt{tl}}) \\ \tilde{\pi}_u^{\mathtt{tl}} & \text{if } c_1(\tilde{\pi}_u^{\mathtt{tl}}) < c_1(\pi_u^{\mathtt{tl}}), \end{cases}$$

and

$$\hat{\pi}_u^{\mathtt{br}} = \begin{cases} \pi_u^{\mathtt{br}} & \text{if } c_2(\pi_u^{\mathtt{br}}) \le c_2(\tilde{\pi}_u^{\mathtt{br}}) \\ \tilde{\pi}_u^{\mathtt{br}} & \text{if } c_2(\tilde{\pi}_u^{\mathtt{br}}) < c_2(\pi_u^{\mathtt{br}}). \end{cases}$$

For a visualization, see Fig. 3.

We are finally ready to describe PP-A$^*$, our algorithm for bi-criteria approximate shortest-path computation (Problem 2). We run a best-first search similar to Alg. 1 but nodes are path pairs. We start with the trivial path pair $(v_{\text{start}}, v_{\text{start}})$ and describe our algorithm by detailing the different functions highlighted in Alg. 1. For each function, we describe what needs to be performed and how this can be efficiently implemented when consistent heuristics are used (see Sec. 3). Finally, the pseudocode of the algorithm is provided in Alg. 2 with the efficient implementations provided in Alg. 3-6.

**Ordering nodes in OPEN:** Recall that a node is a path pair $(\pi_u^{\mathtt{tl}}, \pi_u^{\mathtt{br}})$ and that each path $\pi$ has two $f$ values which correspond to the two cost functions and the two heuristic functions. Nodes are ordered lexicographically according to

$$\left(f_1(\pi_u^{\mathtt{tl}}), f_2(\pi_u^{\mathtt{br}})\right). \tag{3}$$

**Domination checks:** Recall that there are two types of domination checks that we wish to perform (i) checking if

---

**Algorithm 2** PP-A$^*$

**Input:** $(G = (V, E), v_{\text{start}}, v_{\text{goal}}, c_1, c_2, h_1, h_2, \varepsilon_1, \varepsilon_2)$

1: solutions_pp $\leftarrow \emptyset$          ▷ path pairs
2: OPEN $\leftarrow$ new path pair $(v_{\text{start}}, v_{\text{start}})$
3: **while** OPEN $\ne \emptyset$ **do**
4:     $(\pi_u^{\mathtt{tl}}, \pi_u^{\mathtt{br}}) \leftarrow$ OPEN.**extract_min**()
5:     **if is_dominated_PP-A**$^*(\pi_u^{\mathtt{tl}}, \pi_u^{\mathtt{br}})$ **then**
6:        **continue**
7:     **if** $u = v_{\text{goal}}$ **then**       ▷ reached goal
8:        **merge_to_solutions_PP-A**$^*(\pi_u^{\mathtt{tl}}, \pi_u^{\mathtt{br}}, \text{solutions\_pp})$
9:        **continue**
10:    **for** $e = (u, v) \in$ neighbors$(s(n), G)$ **do**
11:      $(\pi_v^{\mathtt{tl}}, \pi_v^{\mathtt{br}}) \leftarrow$ **extend_PP-A**$^*((\pi_u^{\mathtt{tl}}, \pi_u^{\mathtt{br}}), e)$
12:      **if is_dominated_PP-A**$^*(\pi_v^{\mathtt{tl}}, \pi_v^{\mathtt{br}})$ **then**
13:        **continue**
14:      **insert_PP-A**$^*((\pi_v^{\mathtt{tl}}, \pi_v^{\mathtt{br}}), \text{OPEN})$
15: solutions $\leftarrow \emptyset$
16: **for** $(\pi_{v_{\text{goal}}}^{\mathtt{tl}}, \pi_{v_{\text{goal}}}^{\mathtt{br}}) \in$ solutions_pp **do**
17:    solutions $\leftarrow$ solutions $\cup \{\pi_{v_{\text{goal}}}^{\mathtt{tl}}\}$
18: **return** solutions

---

a node is dominated by a node that was already expanded and (ii) checking if a node has the potential to reach the goal with a solution whose cost is not dominated by any existing solution.

In our setting a path pair $\text{PP}_u$ is dominated by another path pair $\tilde{\text{PP}}_u$ if the partial Pareto frontier represented by $\text{PP}_u$ is contained in the partial Pareto frontier represented by $\tilde{\text{PP}}_u$ (see Fig. 4). We can efficiently test if $\text{PP}_u = (\pi_u^{\mathtt{tl}}, \pi_u^{\mathtt{br}})$ is dominated by any path to $u$ found so far, by checking if

$$g_2(\pi_u^{\mathtt{br}}) \ge g_2^{\min}(u). \tag{4}$$

This only holds when using the assumption that our heuristic functions are admissible and consistent and using the way

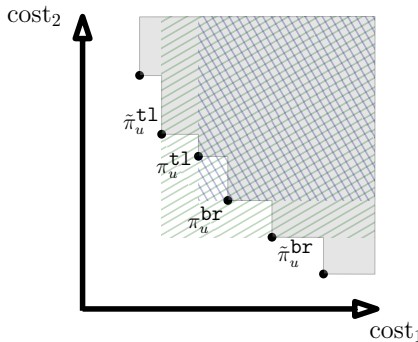

Figure 4: Testing dominance of partial Pareto frontiers using path pairs. The partial Pareto frontier $\Pi_u^{\pi_u^{\mathrm{tl}}, \pi_u^{\mathrm{br}}}$ is contained in the partial Pareto frontier $\Pi_u^{\tilde{\pi}_u^{\mathrm{tl}}, \tilde{\pi}_u^{\mathrm{br}}}$. Thus, the region represented by $\pi_u^{\mathrm{tl}}, \pi_u^{\mathrm{br}}$ is contained in the region represented by $\tilde{\pi}_u^{\mathrm{tl}}, \tilde{\pi}_u^{\mathrm{br}}$.

we order our OPEN list.

We now continue to describe how we test if a path pair has the potential to reach the goal with a solution whose cost is not dominated by any existing solution. Given a path pair $\mathrm{PP}_u = (\pi_u^{\mathrm{tl}}, \pi_u^{\mathrm{br}})$ a lower bound on the partial Pareto frontier at $v_{\mathrm{goal}}$ that can be attained via $\mathrm{PP}_u$ is obtained by adding the heuristic values to the costs of the two paths in $\mathrm{PP}_u$. Namely, we consider two paths $\pi_{v_{\mathrm{goal}}}^{\mathrm{tl}}, \pi_{v_{\mathrm{goal}}}^{\mathrm{br}}$ such that $c_i(\pi_{v_{\mathrm{goal}}}^{\mathrm{tl}}) := c_i(\pi_u^{\mathrm{tl}}) + h_i(u)$ and $c_i(\pi_{v_{\mathrm{goal}}}^{\mathrm{br}}) := c_i(\pi_u^{\mathrm{br}}) + h_i(u)$. Note that these paths may not be attainable and are a lower bound on the partial Pareto frontier that can be obtained via $\mathrm{PP}_u$. Now, if the partial Pareto frontier $\mathrm{PPF}_{v_{\mathrm{goal}}}^{\pi_{v_{\mathrm{goal}}}^{\mathrm{tl}}, \pi_{v_{\mathrm{goal}}}^{\mathrm{br}}}$ is contained in the union of the currently-computed partial Pareto frontiers at $v_{\mathrm{goal}}$, then $\mathrm{PP}_u$ is dominated. Similar to the previous dominance check, this can be efficiently implemented by testing if

$$(1 + \varepsilon_2) \cdot (f_2(\pi_u^{\mathrm{br}})) \geq g_2^{\min}(v_{\mathrm{goal}}). \qquad (5)$$

**Inserting nodes in OPEN:** Recall that we want to use the notion of path pairs to represent a partial Pareto frontier. Key to the efficiency of our algorithm is to have every partial Pareto frontier as large as possible under the constraint that they are all $(\varepsilon_1, \varepsilon_2)$-bounded. Thus, when coming to insert a path pair $\mathrm{PP}_u$ into the OPEN list, we check if there exists a path pair $\tilde{\mathrm{PP}}_u$ such that $\mathrm{PP}_u$ and $\tilde{\mathrm{PP}}_u$ can be merged and the resultant path pair is still $(\varepsilon_1, \varepsilon_2)$-bounded.

If this is the case, we remove $\tilde{\mathrm{PP}}_u$ and replace it with the merged path pair.

**Merging solutions:** Since we want to minimize the number of path pairs representing $\Pi_{v_{\mathrm{goal}}}(\varepsilon_1, \varepsilon_2)$ we suggest an optimization that operates similarly to node insertions. When a new path pair $\mathrm{PP}_{v_{\mathrm{goal}}}$ representing a partial Pareto frontier at $v_{\mathrm{goal}}$ is obtained, we test if there exists a path pair in the solution set $\tilde{\mathrm{PP}}_{v_{\mathrm{goal}}}$ such that $\mathrm{PP}_u$ and $\tilde{\mathrm{PP}}_{v_{\mathrm{goal}}}$ can be merged and the resultant path pair is still $(\varepsilon_1, \varepsilon_2)$-bounded.

---

**Algorithm 3** is_dominated_PP-A*

**Input:** $(\mathrm{PP}_u = (\pi_u^{\mathrm{tl}}, \pi_u^{\mathrm{br}}))$
1: **if** $(1 + \varepsilon_2) \cdot f_2(\pi_u^{\mathrm{br}}) \geq g_2^{\min}(v_{\mathrm{goal}})$ **then**
2:     **return** true     ▷ dominated by solution
3: **if** $g_2(\pi_u^{\mathrm{br}}) \geq g_2^{\min}(u)$ **then**
4:     **return** true   ▷ dominated by existing path pair
5: **return** false

---

**Algorithm 4** extend_PP-A*

**Input:** $(\mathrm{PP}_u = (\pi_u^{\mathrm{tl}}, \pi_u^{\mathrm{br}}), \mathrm{e} = (\mathrm{u}, \mathrm{v}))$
1: $\pi_v^{\mathrm{tl}} \leftarrow$ **extend**$(\pi_u^{\mathrm{tl}})$
2: $\pi_v^{\mathrm{br}} \leftarrow$ **extend**$(\pi_u^{\mathrm{br}})$
3: **return** $(\pi_v^{\mathrm{tl}}, \pi_v^{\mathrm{br}})$

---

**Algorithm 5** insert_PP-A*

**Input:** $(\mathrm{PP}_v, \mathrm{OPEN})$
1: **for each** path pair $\tilde{\mathrm{PP}}_v \in \mathrm{OPEN}$ **do**
2:     $\mathrm{PP}_v^{\mathrm{merged}} \leftarrow \mathrm{merge}(\tilde{\mathrm{PP}}_v, \mathrm{PP}_v)$
3:     **if** $\mathrm{PP}_v^{\mathrm{merged}}$.is_bounded$(\varepsilon_1, \varepsilon_2)$ **then**
4:         OPEN.remove$(\tilde{\mathrm{PP}}_v)$   ▷ remove existing path pair
5:         OPEN.**insert**$(\mathrm{PP}_v^{\mathrm{merged}})$
6:         **return**
7: OPEN.**insert**$(\mathrm{PP}_v)$
8: **return**

---

**Algorithm 6** merge_to_solutions_PP-A*

**Input:** $(\mathrm{PP}_{v_{\mathrm{goal}}}, \mathrm{solutions\_pp})$
1: **for each** path pair $\tilde{\mathrm{PP}}_{v_{\mathrm{goal}}} \in \mathrm{solutions\_pp}$ **do**
2:     $\mathrm{PP}_{v_{\mathrm{goal}}}^{\mathrm{merged}} \leftarrow \mathrm{merge}(\tilde{\mathrm{PP}}_{v_{\mathrm{goal}}}, \mathrm{PP}_v)$
3:     **if** $\mathrm{PP}_{v_{\mathrm{goal}}}^{\mathrm{merged}}$.is_bounded$(\varepsilon_1, \varepsilon_2)$ **then**
4:         solutions_pp.remove$(\tilde{\mathrm{PP}}_{v_{\mathrm{goal}}})$
5:         solutions_pp.insert$(\mathrm{PP}_{v_{\mathrm{goal}}}^{\mathrm{merged}})$
6:         **return**
7: solutions_pp.insert$(\mathrm{PP}_{v_{\mathrm{goal}}})$
8: **return**

---

If this is the case, we remove $\tilde{\mathrm{PP}}_{v_{\mathrm{goal}}}$ and replace it with the merged path pair.

**Returning solutions:** Recall that our algorithm stores solutions as path pairs and not individual paths. To return an approximate Pareto frontier, we simply return one path in each path pair. Here, we arbitrarily choose to return $\pi_{v_{\mathrm{goal}}}^{\mathrm{tl}}$ for each path pair $(\pi_{v_{\mathrm{goal}}}^{\mathrm{tl}}, \pi_{v_{\mathrm{goal}}}^{\mathrm{br}})$.

### 4.3 Analysis

Showing that PP-A* indeed computes an approximate Pareto frontier using the domination checks suggested in Eq. 4 and 5, may be done by using similar arguments as those presented in (Ulloay et al. 2020). However, such a proof is omitted due to lack of space and we refer the reader

| New York City (NY) | | | | | | | |
| 264,346 states, 730,100 edges | | | | | | | |
| $\varepsilon$ | avg $n_{sol}$ | | avg t | | min t | | max t | |
| | PP-A* | BOA* | PP-A* | BOA* | PP-A* | BOA* | PP-A* | BOA* |
|---|---|---|---|---|---|---|---|---|
| **0** | 158 | 158 | 1,047 | 405 | 2 | 0 | 13,563 | 5,038 |
| **0.01** | 19 | 20 | 291 | 353 | 3 | 0 | 3,662 | 4,577 |
| **0.025** | 10 | 10 | 168 | 295 | 2 | 0 | 2,207 | 4,101 |
| **0.05** | 6 | 6 | 111 | 240 | 3 | 0 | 1,523 | 3,538 |
| **0.1** | 4 | 4 | 69 | 174 | 2 | 0 | 932 | 2,694 |
| San Francisco Bay (BAY) | | | | | | | |
| 321,270 states, 794,830 edges | | | | | | | |
| $\varepsilon$ | avg $n_{sol}$ | | avg t | | min t | | max t | |
| | PP-A* | BOA* | PP-A* | BOA* | PP-A* | BOA* | PP-A* | BOA* |
| **0** | 117 | 117 | 1,213 | 423 | 3 | 0 | 21,751 | 7,584 |
| **0.01** | 16 | 17 | 222 | 369 | 4 | 0 | 2,927 | 6,805 |
| **0.025** | 9 | 9 | 127 | 321 | 3 | 0 | 1,530 | 5,614 |
| **0.05** | 5 | 6 | 85 | 272 | 3 | 0 | 1,109 | 4,570 |
| **0.1** | 3 | 4 | 54 | 199 | 3 | 0 | 576 | 3,056 |
| Colorado (COL) | | | | | | | |
| 435,666 states, 1,042,400 edges | | | | | | | |
| $\varepsilon$ | avg $n_{sol}$ | | avg t | | min t | | max t | |
| | PP-A* | BOA* | PP-A* | BOA* | PP-A* | BOA* | PP-A* | BOA* |
| **0** | 318 | 318 | 3,368 | 1,144 | 5 | 1 | 56,153 | 17,348 |
| **0.01** | 15 | 16 | 372 | 944 | 5 | 1 | 3,633 | 16,304 |
| **0.025** | 7 | 8 | 192 | 768 | 5 | 1 | 1,690 | 15,037 |
| **0.05** | 4 | 5 | 116 | 608 | 5 | 1 | 991 | 13,718 |
| **0.1** | 3 | 3 | 69 | 470 | 4 | 1 | 593 | 11,977 |
| Florida (FL) | | | | | | | |
| 1,070,376 states, 2,712,798 edges | | | | | | | |
| $\varepsilon$ | avg $n_{sol}$ | | avg t | | min t | | max t | |
| | PP-A* | BOA* | PP-A* | BOA* | PP-A* | BOA* | PP-A* | BOA* |
| **0** | 357 | 357 | 12,177 | 3,545 | 12 | 3 | 270,450 | 68,467 |
| **0.01** | 12 | 13 | 1,000 | 3,228 | 12 | 3 | 17,092 | 64,642 |
| **0.025** | 6 | 6 | 479 | 2,738 | 11 | 3 | 8,060 | 59,908 |
| **0.05** | 3 | 4 | 263 | 1,985 | 12 | 3 | 3,945 | 39,214 |
| **0.1** | 2 | 2 | 144 | 1,172 | 11 | 2 | 1,780 | 21,665 |

Table 1: Average number of solutions ($n_{\text{sol}}$) and runtime (in ms) comparing $\mathsf{BOA}^*_\varepsilon$ and $\mathsf{PP\text{-}A}^*$ on 50 random queries sampled for four different roadmaps for different approximation factors.

to the extended version of this text (Salzman 2020).

# 5 Evaluation

**Experimental setup.** To evaluate our approach we compare it to $\mathsf{BOA}^*_\varepsilon$ as $\mathsf{BOA}^*$ was recently shown to dramatically outperform other state-of-the-art algorithms for bicriteria shortest path (Ulloay et al. 2020). All experiments were run on an 1.8GHz Intel(R) Core(TM) i7-8565U CPU Windows 10 machine with 16GB of RAM. All algorithm implementations were in C++. We use road maps from the 9'th DIMACS Implementation Challenge: Shortest Path[4]. The cost components represent travel distances ($c_1$) and times ($c_2$). The heuristic values are the exact travel distances and times to the goal state, computed with Dijkstra's algorithm. Since all algorithms use the same heuristic values, heuristic-computation times are omitted.

**General comparison.** Similar to the experiments of Ulloa et al (2020) we start by comparing the algorithms for four different roadmaps containing between roughly $250K$ and $1M$ vertices. Table 1 summarizes the number of solutions in the approximate Pareto frontier and average, minimum and maximum running times of the two algorithms using

[4]http://users.diag.uniroma1.it/challenge9/download.shtml.

| North East (NE) | | | | | |
| 1,524,453 states, 3,897,636 edges | | | | | |
| $\varepsilon$ | avg t | | min t | | max t | |
| | PP-A* | BOA* | PP-A* | BOA* | PP-A* | BOA* |
|---|---|---|---|---|---|---|
| **0** | 192.6 | 59.5 | 0.04 | 0.02 | 2,4189.9 | 592.6 |
| **0.01** | 13.1 | 68.3 | 0.03 | 0.01 | 111.6 | 600.9 |
| **0.025** | 5.6 | 57.3 | 0.02 | 0.01 | 46.9 | 510.9 |
| **0.05** | 2.7 | 40.8 | 0.02 | 0.01 | 22.6 | 345.1 |
| **0.1** | 1.3 | 25.8 | 0.02 | 0.01 | 9.0 | 229.8 |

Table 2: Runtime (in seconds) comparing $\mathsf{BOA}^*$ and $\mathsf{PP\text{-}A}^*$ on 50 random queries sampled for the NE map.

the following values[5] $\varepsilon \in \{0, 0.01, 0.025, 0.05, 0.1\}$. Here, approximation values of zero and $0.01$ correspond to computing the entire Pareto frontier and approximating it using a value of $1\%$, respectively.

When computing the entire Pareto frontier $\mathsf{BOA}^*$ is roughly three times faster than $\mathsf{PP\text{-}A}^*$ on average. This is to be expected as $\mathsf{PP\text{-}A}^*$ stores for each element in the priority queue two paths and requires more computationally-demanding operations. As the approximation factor is increased, the average running time of $\mathsf{PP\text{-}A}^*$ drops faster, when compared to $\mathsf{BOA}^*_\varepsilon$ and we observe a significant average speedup. Interestingly, when looking at the minimal running time, $\mathsf{BOA}^*_\varepsilon$ significantly outperforms $\mathsf{PP\text{-}A}^*$. This is because in such settings the approximate Pareto frontier contains one solution, which $\mathsf{BOA}^*_\varepsilon$ is able to compute very fast. Other nodes are approximately dominated by this solution and the algorithm can terminate very quickly. $\mathsf{PP\text{-}A}^*$, on the other hand, still performs merge operations which incur a computational overhead. When looking at the maximal running time, we can see an opposite trend where $\mathsf{PP\text{-}A}^*$ outperforms $\mathsf{BOA}^*_\varepsilon$ by a large factor.

**Pinpointing the performance differences between $\mathsf{PP\text{-}A}^*$ and $\mathsf{BOA}^*_\varepsilon$.** The first set of results suggest that as the problem becomes harder, the speedup that $\mathsf{PP\text{-}A}^*$ may offer becomes more pronounced. We empirically quantify this claim by moving to a larger map called the North East (NE) map which contains 1,524,453 states and 3,897,636 edges where we obtain even larger speedups (see Table 2).

We plot both the number of nodes expanded (which typically is proportional to running time of $\mathsf{A}^*$-like algorithms) as well as the running time of each algorithm as a function of the approximation factor (see, Fig. 5a and 5b, respectively). Here we used $\varepsilon \in \{0, 0.01, 0.025, 0.05, 0.1, 0.25, 0.5, 1\}$.

We observe that the number of nodes expanded monotonically decreases when the approximation factor is increased for both algorithms. This is because additional nodes may be pruned which in turn, prunes all nodes in their subtree. It is important to discuss *how* these nodes are pruned: Recall that $\mathsf{BOA}^*_\varepsilon$ prunes nodes according to Eq. 2. Thus, increasing the approximation factor only allows to prune more nodes ac-

[5]While $\mathsf{PP\text{-}A}^*$ allows a user to specify two approximation factors corresponding to the two cost functions, this is not the case for $\mathsf{BOA}^*$. Thus, in all experiments we use a single approximation factor $\varepsilon$ and set $\varepsilon_1 = \varepsilon_2 = \varepsilon$.

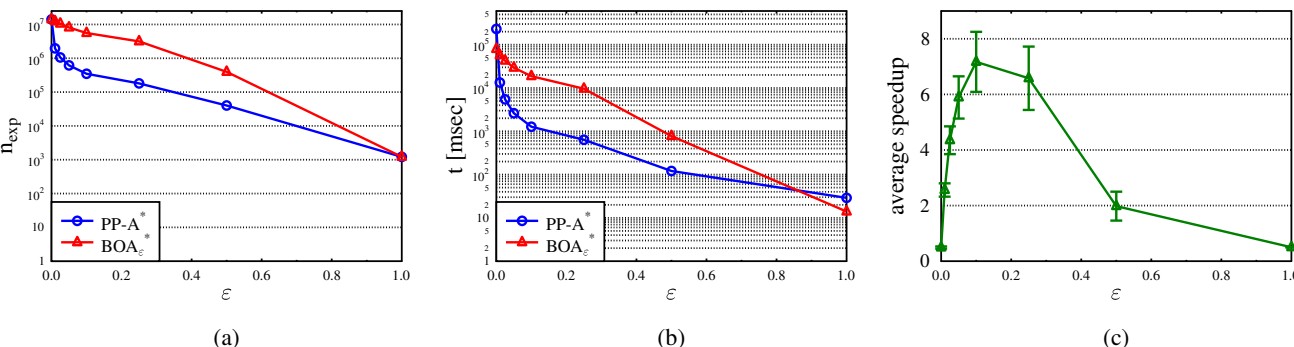

Figure 5: North East (NE) plots. (a) The average number of expanded nodes ($n_{\exp}$) and (b) the average time for both algorithms as a function of the approximation factor. Notice the logarithmic scale in the $y$-axis for both plots. (c) The average speedup of PP-A$^*$ when compared to BOA$^*_\varepsilon$ as a function of the approximation factor. Error bars denote one standard error (error bars in (a) and (b) are not visible due to the logarithmic scale).

cording to the already-computed solutions and not according to the paths computed to intermediate nodes. In contrast, PP-A$^*$ prunes nodes according to Eq. 4 and 5. Thus, in addition to more path pairs being merged, increasing the approximation allows to prune more path pairs according to the already-computed solutions as well as the path pairs computed to intermediate vertices. Thus, for relatively-small approximation factors that are greater than zero (in our setting, $0 < \varepsilon < 0.5$, we see that BOA$^*$ expands a significantly higher number of nodes than PP-A$^*$ which explains the speedups we observed. However, for large approximation factors, there is typically only one solution in the approximate Pareto frontier. This solution, which is found quickly by BOA$^*_\varepsilon$, allows to prune almost all other paths which results in BOA$^*_\varepsilon$ running much faster than PP-A$^*$. This trend is visualized in Fig. 5c.

## 6   Future research

### 6.1   Bidirectional search

We presented PP-A$^*$ as a unidirectional search algorithm, however a common approach to speed up search algorithms is to perform two simultaneous searches: a forward search from $v_{\text{start}}$ to $v_{\text{goal}}$ and a backward search from $v_{\text{goal}}$ to $v_{\text{start}}$ (Pohl 1971). Thus, an immediate task for future research is to suggest a bidirectional extension of PP-A$^*$. Here we can build upon recent progress in bi-directional search algorithms for bi-criteria shortest-path problems (Sedeño-Noda and Colebrook 2019).

### 6.2   Beyond two optimization criteria

We presented PP-A$^*$ as a search algorithm for two optimization criteria, however the same concepts can be used for multi-criteria optimization problems. Unfortunately, it is not clear how to perform operations such as dominance checks efficiently since the methods presented for BOA$^*$ do not extend to such settings.

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
