# OpenReview forum: "Approximate bi-criteria search by efficient representation of subsets of the Pareto-optimal frontier"
_icaps-conference.org/ICAPS/2020/Workshop/HSDIP — HSDIP 2020_

### Official Review · AnonReviewer2 · 2020-08-17
**An interesting new algorithm for sub-optimal bi-objective search.**

**Rating:** 7
**Confidence:** 3

**Review:**

This paper considers search problems with two cost functions. The solution is a set of
paths that form a pareto frontier according to both cost functions.

Even though the evaluation is performed in a single domain, I consider the exploration of
search algorithms in the scope of the workshop. The algorithm is quite interesting, by
considering pairs of paths in each node, is able to better take advantage of the sub
optimality factor epsilon.

The paper is well-written. There are only a couple of things that I think should be
changed for the final version:

 - I disagree with the definition of strict dominance introduced in Definition 1, because
it requires both costs be strictly better, and it should suffice only one of them. The
definition of strict dominance that I'd suggest for solutions x and y is (a) there is weak
dominance, and (b) c1(x) < c1(y) or c2(x) < c2(y).

According to Definition 1, (10, 10) is not strictly dominated by (10, 5). This has an
impact on the definition of pareto frontier. For example, with the definition in the paper
{(10, 0), (10, 1), (10, 2), (10, 3), (10, 4), (5, 5),} would be a valid frontier, whereas
in reality it should only be {(10, 0), (5, 5),}. This would significantly increase the
size of the pareto-optimal frontier. I assume that the implementation does not allow for
this and indeed performs the check I suggest above. Nevertheless, it'd be great if you can
comment on this on the rebuttal/discussion.

- A solution set whose size is often exponential in the size of the input -> I don't think
this is correct. The solution set is exponential in the number of cost functions, but not
in the size of your input. Perhaps one can construct examples where this is true, but
definitively not with correlated cost functions like time and distance on a road map. In
fact, this is highlighted in the experiments, where larger maps have larger solution sets
but I don't think this increases by more than a linear factor.

The experimental evaluation is convincing in showing the advantage of the new approach
over the state of the art BOA*. Some questions/comments about the evaluation:

- It is mentioned that the source code is in C++. Was BOA* reimplemented from scratch?  Is
  the code available somewhere? You could consider publishing the code and experimental
  data in a public archive like Zenodo.

- I'm not a fan of 3D plots since I myself can hardly distinguish where the points are
  exactly located. I'd suggest to use only a 2D scatter plot and use different colors for
  the third axis.

- You only consider BOA* as an alternative. Have you considered other algorithms of the literature? How does the new algorithm compare against algorithms that find the exact solution when epsilon is 0?

I personally would change the section on Future research by a Conclusions section. Future
research directions can still be mentioned there, but by removing the two sub-section
headers you can also add a short summary of the main conclusions the reader should have
drawn from this paper.

Minor comments:

p1. then the others -> than the others
p2.  the space of solution*s*
p2. approxiamtion -> approximation
p2. *c*computing
p3. In Problem 1, I found the sentence "The bi-criteria shortest path calls for ..." hard to read. I suggest using "The bi-criteria shortest path problem calls for ...", and same for problem 2.
p4. "With the risk of being tedious", I'd personally remove that sentence since it does not really add anything and, if something, makes the reader less interested in the upcoming section.
p4. I was first confused with Equation (1), perhaps it could be highlighted that one does not need to compare the solutions according to the first cost funciton because nodes in the open list are sorted according to such a cost function. Also, I guess this only works if the heuristic is consistent, os perhaps that should also be clarified.
p5. interediate -> intermediate
p6. simply correspond*s*
p6. when consistent heuristic*s* are
p6 f values, correspond -> which correspond or corresponding

---

> ### Author Response · Authors · 2020-08-31
> **Response to AnonReviewer2**
>
> Thank you for your helpful comments and the positive review. Regarding your comments:
>
> (1) Definition of strict dominance: The reviewer is correct and we thank him for pointing this out. Indeed the implementation follows the reviewer's suggested definition and Definition (1) will be updated accordingly.
>
> (2) Exponential size of Pareto frontier: The statement in the paper is indeed correct and a specific example was constructed by Dollevoet et al. The reviewer is correct that this holds for general cost functions and not "correlated" cost functions. We will update the text to clarify this point.
>
> (3) Code implementation: BOA* was implemented from scratch. The code is going through additional optimization and will be publicly available after this stage is completed.
>
> (4) Comparison with alternative algorithms: Since BOA* was shown to be orders of magnitude faster than other alternatives and existing approximate algorithms do not offer significant speedups in practice (see, [Breugem, Dollevoet, and van den Heuvel 2017]) the approach was only compared with BOA*.

---

### Official Review · AnonReviewer1 · 2020-08-18

**Rating:** 7
**Confidence:** 3

**Review:**

This paper addresses planning problems with two objective functions to optimize for. In general, there is no solution that minimizes the two functions, and so Pareto-optimal solutions are sought. The paper presents an algorithm for computing an approximation of the Pareto-optimal frontier, and evaluates their performance.

The paper is relevant to the topic of the workshop, and contributes towards advancing the state of the art with algorithms and experimental results.

The paper is in general well written. Perhaps a weak point in the paper, as it is currently presented, is that there is a lot of background information to cover until the algorithmic description is introduced in section 4.2. The authors seem to acknowledge that (e.g., with a related comment in the beginning of section 3). I do not have a good solution, but maybe the contents of Section 3 can be presented together with the description of the algorithm. I suggest the authors rethink the organization of the paper should they plan on re-submitting it to another venue as a full paper.

In your experiments, have you tried with different heuristics? If so, what is (or you think will be) the performance of PP-A* relative to BOA*, and their sensitiveness to non-exact consistent heuristics?

Minor comments:
The paper submission was not anonymized.
Equation (1): I believe there is a typo and f_2 should be g_2
Table 1: there is a lot of information to digest. I would have found it useful to mark in bold the results of the best configuration. Perhaps a column that shows the percentage (%) of improvement of PP-A* would also improve the interpretability of the results.
Section 4.3 could be merged with the paragraph above in Section 4.2.  The subsections in Section 6 are very short, and could be replaced with paragraphs.
Figure 6: I suggest the authors to replace the 3D plot with two X-Y plots that show, e.g., the speedup ratio vs. the number of solutions (rep., number of expanded nodes).

---

> ### Author Response · Authors · 2020-08-31
> **Response to AnonReviewer1**
>
> Thank you for your helpful comments and the positive review.
> Regarding your comments:
>
> (1) Experimenting with different heuristics: As mentioned in the paper, both BOA*, and subsequently PP-A* can only perform efficient dominance checks when the heuristic is consistent. However, the algorithmic approach of PP-A* does not rely on this assumption. 	In such cases, a new evaluation comparing PP-A* with other state-of-the-art algorithms such as NAMOA* is required but such extensions are left for future work.
>
> (2) Possible typo in Eq. (1): There is no typo here. The second part of Eq. (1) checks if the estimated cost of a solution (hence the f-value and *not* the g-value) is dominated by an already-found solution.
>
> (3) Table (1): The presentation of data in Table (1) was chosen for ease of comparison with the original BOA* paper which included the exact same tables. The NE plots (Fig. 5) are complementary and convey the data the reviewer requested.

---

### Comment · Program_Chairs · 2020-09-14
**Final Decision: Accept**

Dear Authors,

Thank you very much for your submission. We are happy to inform you that we have decided to accept it and we look forward to your talk in the workshop. You will receive additional information per mail in the coming days.

Best,
The HSDIP'20 team

---

### Decision · Program_Chairs · 2020-09-30

Accept